# Investigating the Endo-Lysosomal System in Major Neurocognitive Disorders Due to Alzheimer’s Disease, Frontotemporal Lobar Degeneration and Lewy Body Disease: Evidence for SORL1 as a Cross-Disease Gene

**DOI:** 10.3390/ijms222413633

**Published:** 2021-12-20

**Authors:** Luisa Benussi, Antonio Longobardi, Cemile Kocoglu, Matteo Carrara, Sonia Bellini, Clarissa Ferrari, Roland Nicsanu, Claudia Saraceno, Cristian Bonvicini, Silvia Fostinelli, Roberta Zanardini, Marcella Catania, Matthieu Moisse, Philip Van Damme, Giuseppe Di Fede, Giuliano Binetti, Christine Van Broeckhoven, Julie van der Zee, Roberta Ghidoni

**Affiliations:** 1Molecular Markers Laboratory, IRCCS Istituto Centro San Giovanni di Dio Fatebenefratelli, I-25125 Brescia, Italy; lbenussi@fatebenefratelli.eu (L.B.); alongobardi@fatebenefratelli.eu (A.L.); sbellini@fatebenefratelli.eu (S.B.); rnicsanu@fatebenefratelli.eu (R.N.); csaraceno@fatebenefratelli.eu (C.S.); cbonvicini@fatebenefratelli.eu (C.B.); rzanardini@fatebenefratelli.eu (R.Z.); 2Neurodegenerative Brain Diseases, VIB Center for Molecular Neurology, VIB, B-2610 Antwerp, Belgium; Cemile.Kocoglu@uantwerpen.vib.be (C.K.); christine.vanbroeckhoven@uantwerpen.vib.be (C.V.B.); Julie.vanderZee@uantwerpen.vib.be (J.v.d.Z.); 3Department of Biomedical Sciences, University of Antwerp, B-2000 Antwerp, Belgium; 4Service of Statistics, IRCCS Istituto Centro San Giovanni di Dio Fatebenefratelli, I-25125 Brescia, Italy; carrara.matt@gmail.com (M.C.); cferrari@fatebenefratelli.eu (C.F.); 5MAC-Memory Clinic and Molecular Markers Laboratory, IRCCS Istituto Centro San Giovanni di Dio Fatebenefratelli, I-25125 Brescia, Italy; sfostinelli@fatebenefratelli.eu (S.F.); gbinetti@fatebenefratelli.eu (G.B.); 6Neurology 5/Neuropathology Unit, Fondazione IRCCS Istituto Neurologico Carlo Besta, I-20133 Milan, Italy; Marcella.Catania@istituto-besta.it (M.C.); Giuseppe.DiFede@istituto-besta.it (G.D.F.); 7Department of Neurosciences and Leuven Brain Institute (LBI), KU Leuven-University of Leuven, B-3000 Leuven, Belgium; matthieu.moisse@kuleuven.vib.be (M.M.); philip.vandamme@uzleuven.be (P.V.D.); 8Laboratory of Neurobiology, Center for Brain and Disease Research, VIB, B-3000 Leuven, Belgium; 9Department of Neurology, University Hospitals Leuven, B-3000 Leuven, Belgium

**Keywords:** SORL1, DNAJC6, PPT1, endo-lysosomal genes, NGS, cross-disease, loss of function, multicarrier, allele dose effect

## Abstract

Dysfunctions in the endo-lysosomal system have been hypothesized to underlie neurodegeneration in major neurocognitive disorders due to Alzheimer’s disease (AD), Frontotemporal Lobar Degeneration (FTLD), and Lewy body disease (DLB). The aim of this study is to investigate whether these diseases share genetic variability in the endo-lysosomal pathway. In AD, DLB, and FTLD patients and in controls (948 subjects), we performed a targeted sequencing of the top 50 genes belonging to the endo-lysosomal pathway. Genetic analyses revealed (i) four previously reported disease-associated variants in the SORL1 (p.N1246K, p.N371T, p.D2065V) and DNAJC6 genes (p.M133L) in AD, FTLD, and DLB, extending the previous knowledge attesting SORL1 and DNAJC6 as AD- and PD-related genes, respectively; (ii) three predicted null variants in AD patients in the SORL1 (p.R985X in early onset familial AD, p.R1207X) and PPT1 (p.R48X in early onset familial AD) genes, where loss of function is a known disease mechanism. A single variant and gene burden analysis revealed some nominally significant results of potential interest for SORL1 and DNAJC6 genes. Our data highlight that genes controlling key endo-lysosomal processes (i.e., protein sorting/transport, clathrin-coated vesicle uncoating, lysosomal enzymatic activity regulation) might be involved in AD, FTLD and DLB pathogenesis, thus suggesting an etiological link behind these diseases.

## 1. Introduction

Major neurocognitive disorders due to Alzheimer’s disease (AD), Frontotemporal Lobar Degeneration (FTLD), and Lewy body disease (DLB), are all characterized by abnormal protein accumulation [1,2]. AD is characterized by the deposition of beta-amyloid (Aβ) and phosphorylated tau peptides [3]. DLB, by alpha-synuclein deposits [4], and FTLD presents tau-, ubiquitin-, Fused-in-Sarcoma (FUS)-, and TAR DNA-binding protein 43 (TDP-43)-positive inclusions [5]. The pathogenesis of neurodegenerative disease was recently reviewed, describing AD as a mixed proteinopathy (amyloid and tau) frequently associated with other age-related co-pathologies, such as cerebrovascular lesions, Lewy and TDP-43 pathologies [6]. Moreover, it has been hypothesized that protein aggregates spread from neuron to neuron contributing to the progression of the disease [7]. Exosomes, a specific subtype of extracellular vesicle (EV) of endosomal origin, have been suggested as potential carriers of misfolded toxic proteins: Aβ and tau in AD [8] and alpha-synuclein in Parkinson’s disease (PD)/DLB [9]. 

The endo-lysosomal pathway is essential in maintaining protein homeostasis in the cell. Growing evidence suggests that endosomal and lysosomal dysfunctions, or dysregulation in protein trafficking, play an important role in neurodegeneration, leading to neurocognitive disorders [10]. Numerous human genetics studies support a critical role of endo-lysosomal dysfunction in AD and FTLD. In AD, mutations in presenilin 1 (the most common genetic cause of AD) may, besides altering amyloid processing, result in defective lysosomal acidification and proteolytic activity [11]. In FTLD, mutations in multiple genes related to lysosome and autophagy function have been described, including in GRN, C9orf72, SQSTM1/p62, UBQLN2, DCTN1, TBK1, OPTN, and VCP [12]. Specifically, a progranulin deficiency due to the presence of GRN pathogenic null mutations, one of the most common genetic causes of FTLD, leads to an upregulation of lysosomal genes as well as profound lysosomal defects [13]. Moreover, GRN null mutations cause a strong alteration of the release and composition of exosomes [14]. Of note, homozygous GRN null mutations cause adult-onset neuronal ceroid lipofuscinosis (NCL), a lysosomal storage disorder [15]. A pathogenic expansion in the C9orf72 gene, a common genetic cause of FTLD, also affects lysosomal function. In C9orf72 null mice, by mimicking the reduced C9orf72 expression observed in expansion carriers, a lysosomal enlargement and accumulation was reported [16]. Recently, it has been demonstrated that NDST3, a potent regulator of lysosomal functions, is downregulated in tissues and cells from FTD patients with C9orf72 haploinsufficiency [17]. The link between AD and FTLD and lysosomal genes was also suggested by genome-wide association studies (GWAS). Regarding AD, a pathway enrichment analysis of three large GWAS provided evidence that genetic variation within the endo-lysosomal system is associated with late-onset AD [18]. In FTLD, a large international GWAS revealed a genetic locus linked to the disease risk encompassing the RAB38 and cathepsin C (CTSC) genes, both involved in the lysosomal pathway [19]. In addition, recent large scale GWAS have provided insights into the genetic risk factors associated with PD, showing that the main contributors to PD etiology are, among others, the molecular processes underlying endo-lysosomal dysfunction [20].

Recently, we provided evidence that an alteration in EV release is common in AD, DLB, and FTLD. Specifically, we found a significant reduction in the plasma concentration of EVs and larger sized EVs in all patient groups: these EV parameters together can distinguish patients from controls with a strong sensitivity and specificity [21]. This study further supports the growing body of evidence that endo-lysosomal dysfunctions may be a converging mechanism in neurodegenerative diseases. The aim of this study is to investigate whether AD, DLB and FTLD share genetic variability in the genes involved in the endo-lysosomal pathway. To reach this goal, we performed, in a large group of AD, DLB and FTLD patients, a targeted deep sequencing of the top 50 genes belonging to the endo-lysosomal pathway. The top 50 genes were selected based on a high intolerance to variation and high expression in two or more brain regions (Appendix A). The large majority of the selected genes were described to have a potential role in AD, DLB/PD, FTLD pathogenesis, as reported in human genetics and/or human/mouse molecular studies (Appendix A).

## 2. Results

### 2.1. Identification of Previously Reported Disease-Associated Variants and Predicted Null Variants

Targeted genetic screening for the presence of variants in the coding regions of 50 candidate endo-lysosomal genes was performed on a total of *n* = 697 patients (*n* = 282 AD, *n* = 114 DLB, *n* = 301 FTLD) and *n* = 251 controls (Table 1).

Thirty-one out of the fifty candidate endo-lysosomal genes were described to have a potential role in AD, DLB/PD, and FTLD pathogenesis, as reported in human genetics and human/mouse molecular studies. Specifically, out of the 31 genes, 12 genes were reported in the literature to present risk alleles or mutations associated with AD, DLB/PD and FTLD (Appendix A). In the present study, we detected four previously reported disease-associated variants and three stop-gain heterozygous variants (Table 2, Table 3). Specifically, we found three previously reported variants in the sortilin-related receptor 1 (SORL1) gene: the SORL1 p.N1246K heterozygous variant, previously described as an AD risk factor [22,23], was found in a familial AD patient; the SORL1 p.N371T and p.D2065V heterozygous variants, previously described in two AD patients of North European ancestry [24], were found in an early onset AD and in an FTLD patient and in *n* = 4 AD, *n* = 2 DLB and *n* = 6 FTLD patients, respectively (Table 4). Moreover, in an FTLD and a DLB patient we found a variant in the DnaJ heat shock protein family (Hsp40) member C6/auxilin gene (DNAJC6 p.M133L), previously described in a sporadic early onset PD patient, even if of uncertain significance [25]. The effect of the variants on the protein stability in terms of the ∆∆G was assessed by the Mu-Pro and I-mutant 2.0 tools, and was available for three out of four of the known variants: of note, both computational tools converged in defining these variants as potentially deleterious based on their effect on protein stability. 

In addition to these variants, we found three stop-gain heterozygous variants in SORL1 and palmitoyl-protein thioesterase 1 (PPT1) genes that are predicted to cause a loss of function (LOF) due to haploinsufficiency (Table 3); for both genes, LOF is a known mechanism of the disease. Specifically, loss of function mutations of SORL1 have been described in AD [26,27,28,29]. The LOF SORL1 R985X carrier was an early onset (46 years) familial AD patient and the LOF SORL1 R1207X carrier was a 68-year-old AD patient with an unknown family history. Similarly, we found a stop-gain variant in PPT1 (p.R48X) in an early onset (61 years) familial AD patient; PPT1 loss of function mutations have been previously demonstrated to cause an adult form of neuronal ceroid lipofuscinosis (NCL) [30,31]. 

Considering all the carriers of these variants, the majority of patients (70%) had a disease onset of ≤65 years and/or a positive family history (Table 4). All the previously described disease-associated variants and LOF variants detected in our patients were absent in the controls, and three of the previously described variants were found to be cross-disease variants. Specifically, the SORL1 p.D2065V variant was detected in all three diseases (AD, DLB, FTLD) and was present both in the Italian and Belgian patients; the SORL1 p.N371T variant was found in AD and FTLD; the very rare DNAJC6 p.M133L variant was found in AD and DLB. 

### 2.2. Association Analyses of the Endo-Lysosomal Pathway Genes Involved in Neurodegenerative Diseases and Other Neurological Disorders

Considering all the rare variants detected in the selected 50 genes, including the previously reported variants and the LOF variants, we performed a single variant association analysis in the three diagnostic groups separately (AD, DLB, FTLD vs. CTRL) as well as in all the patients as a whole group (AD + DLB + FTLD vs. CTRL). We found five variants associated with the investigated diseases (Table 5). All these variants were absent in the controls. The effect of the variants on protein stability in terms of the ∆∆G was available for four variants and was assessed by the Mu-Pro or I-mutant 2.0 tool. Both computational tools converged in defining all variants as potentially deleterious based on their effect on protein stability. None of these enrichments were significant after multiple test correction, but we observed some nominally significant results of potential interest, considering the known role of SORL1 in AD [22,23,24,32]. Specifically, the previously AD-associated SORL1 p.D2065V variant was nominally associated with disease, when considering all the patients (*p* = 0.031) as well as in the DLB (*p* = 0.048) and FTLD (*p* = 0.021) groups, separately.

In order to explore whether rare variants were enriched in specific genes, we performed a gene burden analysis and found a nominally significant burden of variants in four genes: SORL1, DNAJC6, NEU1 and AP2A2 (Table 6). Once again, even if none of these enrichments were significant after multiple test correction, we observed some nominally significant results of potential interest, both in SORL1 which was consistently reported to be associated with AD, and in the DNAJC6 gene, consistently reported to be associated with PD [25,35,36]. The SORL1 gene showed a nominally significant burden of variants in the all patients group (*p* value skat = 0.038) and in FTLD patients (p.burden = 0.025 and p.skato = 0.016), and for DNAJC6 there was a variant burden in DLB patients (p.burden = 0.048 and p.skato = 0.048). The localization of variants within the protein sequence and functional domains is reported in Figure 1. In the SORL1 encoded protein (Sortilin-related receptor 1), all patient-specific variants (including the two stop-gain variants) were located in the VPS10, Low-density lipoprotein (LDL) receptor class a (LDL_recept_a), LDL receptor class b (LDL_recept_b) and Fibronectin type III (fn3) domains. Specifically, (i) the known p.N371T, p.N1246K and p.D2065V variants were located in the VPS10, LDL_recept_a and fn3 domains, respectively; (ii) the LOF p.R985X and p.R1207X variants were located in the LDL_recept_b and LDL_recept_a domains, respectively. In the DNAJC6 encoded protein (auxilin) the p.M133L known variant was located in the Tensin-type phosphatase (PTEN) domain, while two additional variants detected in patients were not located in a functional domain.

### 2.3. Multiple Variant Carriers in the Endo-Lysosomal Pathway

Considering previously reported disease associated variants (Table 2) and predicted LOF variants (Table 3), we described four patients carrying an additional rare/very rare variant which was of unknown significance and absent in the controls (Table 7). Specifically (i) the known SORL1 p.D2065V variant was found along with a rare variant in GGA3 in a familial FTLD patient with a disease onset at 68 years; (ii) the LOF SORL1 p.R1207X was found along with a missense variant also in SORL1 (p.D140N) in an AD patient with a disease onset at 68 years and an unknown family history; (iii) the known DNAJC6 p.M133L variant was found along with a rare variant in AGRN (p.A897V) in an AD patient with a disease onset at 65 years and an unknown family history; (iv) the LOF PPT1 p.R48X variant was found along with a rare variant of GNPTG (p.R66Q) in an early onset familial AD patient. In addition, as shown in Table 7, considering the patients with a disease onset of ≤65 years and/or a positive family history, we described 14 more patients carrying two or three rare/very/ultrarare variants which were of unknown significance, and absent in controls. Among these, for example, were three patients carrying the ultrarare ABCA2 p.H1449P variant (found to be nominally associated with disease but this did not survive after multiple test correction) and one early onset (49 years) apparently sporadic FTLD patient who was carrying two compound heterozygous variants in the vacuolar protein sorting-associated protein 52 homolog gene (VPS52 p.Y508C, p.R578W).

## 3. Discussion

Neurodegenerative diseases are characterized by abnormal intracellular protein inclusions or extracellular protein aggregates [1]. There is strong interest in understanding the common molecular mechanisms that contribute to neurodegenerative disorders, and thus, in exploring the etiological link behind these brain diseases [39].

Alterations in the endo-lysosomal system, leading to the failure of proper protein trafficking and degradation, have been hypothesized to underlie neuronal dysfunction in these diseases. Emerging data argue for an interdependence between the production of exosomes (a specific subtype of EVs of endosomal origin) and endosomal pathway integrity in the brain [40]. We recently described that an alteration in the release of EVs is common across AD, FTLD, and DLB and that plasma EV parameters (EV concentration/size) can distinguish patients from controls with strong sensitivity and specificity [21]. Evidence from monogenic diseases and experimental models suggest that autophagy and the endo-lysosomal system may be mechanistically involved in the neurodegenerative processes leading to AD and FTLD [41]. Genetic variation within the endo-lysosomal system is associated with a late-onset AD risk, as demonstrated by the pathway enrichment analysis of three large GWAS: of note, this aggregate genetic association was unique for the autophagic and endo-lysosomal system, and in the same study, an association signal was also observed in PD [18].

Herein, we investigated genetic variability in the genes involved in the endo-lysosomal pathway in major neurocognitive disorders. In a large group consisting of AD, DLB, FTLD and CTRL subjects (948 in total), we performed targeted deep sequencing of the top 50 genes belonging to the endo-lysosomal pathway, with prioritization based on a high intolerance to variation and high brain expression (Appendix A). Specifically, since we were looking for cross-disease genetic variants, we selected genes highly expressed in at least two brain regions. Genetic analyses revealed, in our patients’ dataset, four previously described variants in SORL1 and DNAJC6 genes. The SORL1 p.N1246K variant, previously described as an AD risk factor [22,23], was found in a familial AD patient; the SORL1 p.N371T and the SORL1 p.D2065V variants, previously described in two unrelated AD patients of North European ancestry [24], were found in our dataset in AD, DLB and FTLD patients. SORL1 p.N371T was found both in early onset AD and FTLD, while SORL1 p.D2065V was found in several patients from all three diagnostic groups (four AD, two DLB and six FTLD patients). The DNAJC6 p.M133L variant was found in an AD patient and a DLB patient. This variant was previously described in a sporadic early onset PD patient [25], and even if of uncertain significance, our data also suggest a role of this variant in AD and DLB, an alpha-synuclein associated disease like PD. In addition to known variants, predicted null variants in SORL1 and PPT1 were found in familial early onset AD cases (SORL1 p.R985X, and PPT1 p.R48X) and in an AD patient with an unknown family history (SORL1 p.R1207X). Loss of function variants of SORL1 have already been described in AD [26,27,28,29] and were proposed to cause AD by inducing defects in the endolysosome-autophagy network [42]. PPT1 loss of function variants, in the homozygous or compound heterozygous state, were demonstrated to cause neuronal ceroid lipofuscinosis, an inherited, progressive neurodegenerative disease [30,43].

Interestingly, one of the most common worldwide progranulin loss of function mutations is associated with FTLD in the heterozygous state and with NCL in the homozygous state [15,44]. Similarly, we cannot exclude strictly different clinico-pathological phenotypes (AD versus NCL) also determined by the PPT1 mutation dosage. 

Protein palmitoylation is an important process to regulate the physiological function of the brain. A number of studies have reported that defects in the palmitoylation step or in the enzymes for palmitoylation/depalmitoylation are associated with several neurological disorders including AD [45]. Since APP palmitoylation seems to enhance the amyloidogenic pathway, the loss of a depalmitoylating enzyme such as PPT1 might result in an increased amyloid production; this hypothesis is line with the observed clinical phenotype of the carrier, an early onset familial AD patient. Since LOF is a known mechanism of disease for both genes, the evidence that these variants are pathogenic is strong [46].

Single variant association tests further supported a role of the SORL1 p.D2065V variant as a genetic determinant/risk factor for AD, DLB and FTLD. The gene burden analysis showed a nominally significant burden of potential interest in SORL1 in the all patients group and in FTLD patients, and in DNAJC6 in the DLB group. Regarding SORL1, the majority of the variants found in patients were located in the VPS10 protein domain, the Low-density LDL_recept_a/b domains and the fibronectin type III domain. Of note, a recent meta-analysis of burden tests at the protein domain level of SORL1 missense variants showed a significant association of the VPS10, LDL_recept_a and fibronectin type III domains with AD [47]. The LDL_recept_a domain was demonstrated to have a critical role for the function of SORL1 in AD, as it is involved in amyloid precursor protein (APP) binding and APP retrograde endosome transport to the TGN [48]. The VPS10 domain is involved in Aβ peptides binding and targeting to the lysosomes and variants in this domain have been described which impair lysosomal sorting of Aβ and cause familial AD [27], and the fibronectin type III domain interacts with APP [49]. More importantly, we also provided evidence of an involvement of this gene in FTLD and DLB as we described cross-disease variants, specifically, (i) a previously known AD variant in FTLD and AD patients, and (ii) a previously known AD variant in FTLD, DLB and AD patients, that was found to be a nominally significant associated variant. Regarding the DNAJC6 encoded protein auxilin, only the p.M133L variant was located in a functional domain, and specifically, in the PTEN domain, which is important for the recruitment of auxilin onto clathrin-coated vesicles. Auxilin has a well-established role in clathrin uncoating [50]. Since endocytosis and clathrin-uncoating defects at synapses were demonstrated in auxilin knockout mice, a specialized role for this protein in the clathrin-dependent recycling of synaptic vesicles at synapses was suggested [51]. Of note, a splicing variant affecting the PTEN domain was associated with juvenile parkinsonism [52]. The present study also suggests the possible involvement of this gene in DLB and AD. Interestingly, some of the patients carrying known variants and predicted LOF variants also carried additional rare/very rare variants of still unknown significance (but not present in controls) in endo-lysosomal genes. In addition, considering the patients with early onset disease and a positive family history, we described 14 more cases carrying two or three rare/very rare/ultrarare variants of still unknown significance (but not present in controls). Among these, (i) a Belgian patient with a very early onset FTLD (49 years) carrying two potentially damaging compound heterozygous variants in VPS52, a subunit of the Golgi-associated retrograde protein complex (interacting with the PD-associated LRRK2) which is involved in retrograde transport of early and late endosomes to the Golgi [53]; (ii) three patients carrying the ultrarare ABCA2 p.H1449P variant, which was found to be nominally associated with disease, but this did not survive after multiple test correction.

There are limitations in the present study: (i) This being a pilot study, further validation in larger groups is needed. Such large studies will, on one side give a definitive answer on the role played by SORL1, DNAJC6, and PPT1 in neurocognitive disorders and, on the other side, unveil the potential of the AP2A2, ABCA2, NEU1, TOM1, and AGRN genes in contributing to the disease. (ii) Disease segregation studies on families for variants with the strongest evidence of pathogenicity are needed. (iii) We explored only a portion of endo-lysosomal genes and thus a more comprehensive genetic screening (including all genes belonging to this pathway) as well functional studies on identified variants could be of interest. 

Based on our data and the literature data on the role played by the endo-lysosomal system in neurocognitive disorders, epigenetic studies evaluating the complex interplay of genetic and environmental factors are warranted to better explore the etiological link behind these diseases. The final goal is to develop new strategies for the development of innovative therapeutic approaches targeting the endo-lysosomal pathway and taking advantage of the current knowledge in this field [54,55,56].

Our data highlight that genes controlling key endo-lysosomal processes such as intracellular protein sorting/transport, the uncoating of clathrin-coated vesicles and the regulation of enzymatic activity in lysosomes, might be involved in AD, DLB and FTLD pathogenesis. Altogether our data further confirm the key role of the endo-lysosomal pathway in these diseases and suggest the existence of cross-diseases mechanisms involved in major neurocognitive disorders. Our data strongly support a critical role for SORL1 in AD and related diseases and highlight SORL1 as a potential therapeutic target for drug development.

## 4. Materials and Methods

### 4.1. Participants

This retrospective study was carried out on DNA from a total of *n* = 697 patients (*n* = 282 AD, *n* = 114 DLB, *n* = 301 FTLD) and *n* = 251 subjects with normal cognitive function (CTRL) (Table 1). Clinical diagnosis for probable AD, DLB and FTLD was made according to international guidelines [57,58,59,60,61,62]. The family history was determined by a family history questionnaire and FTLD pedigrees were classified as previously described [63,64,65]. DNA samples were available from the biological banks of the IRCCS Fatebenefratelli Brescia and IRCCS Besta Milan (Italian cohort), from the Neurodegenerative Brain Diseases Human Biobank of the VIB Center for Molecular Neurology, Antwerp, Belgium (Belgian patient cohort [66,67]), and from Project MinE (Belgian control cohort, [68]). Written informed consent was obtained from all subjects. The study protocol was approved by the local ethics committee (Prot. N. 111/2017).

### 4.2. Gene Selection

The candidate genes were selected employing web resources: KEGG pathway, Gene Ontology (GO), Gene Set Enrichment Analysis (GSEA), and Reactome for selecting genes belonging to the endo-lysosomal pathway. The Genotype-Tissue Expression (GTEx) dataset for gene selection was based on brain expression and the Residual Variation Intolerance Score (RVIS) score was used as a ranking method to prioritize genes with a high intolerance to variation (more negative values express an increasing intolerance to mutations). A list of 314 genes was obtained including genes belonging to pathways connected to lysosomes, endosomes and endo-lysosomes. Genes were then filtered according to brain tissue expression. For each gene, in each region we calculated if it fell above the 3rd quartile of expression for that specific tissue and we filtered out any genes that were not highly expressed in at least two brain regions. Applying this strategy, we selected 93 genes. Fifty genes were further selected from this list, with a priority given to genes with the lowest RVIS score (Appendix A). 

### 4.3. Genetic Analyses

The entire coding regions of the 50 candidate genes were analyzed by amplicon-based target enrichment and Next-Generation Sequencing (NGS) of the exons and exon-intron boundaries on a Illumina^®^ MiSeq platform (Illumina, San Diego, CA, USA). NGS analysis was performed on *n* = 65 AD; *n* = 102 DLB, *n* = 58 FTLD and *n* = 75 CTRL samples. The quality assessment of gDNA was performed on a 0.8% agarose gel and gDNA was quantified with a Qubit dsDNA HS Assay Kit (Thermo Fisher Scientific, Waltham, MA, USA). A total of 200 ng of gDNA was used for library preparation with a Nextera Flex for Enrichment kit (Illumina, Inc., USA). gDNA was tagmented, amplified and purified with AMPure XP Beads (Beckman Coulter, Inc., Brea, CA, USA). The size, quality and quantity of libraries was assessed with a High Sensitivity DNA kit on a Bioanalyzer instrument (Agilent Technologies, Santa Clara, CA, USA). A 12 pM sample of the pooled library was loaded on a MiSeq reagent cartridge v3 and sequenced on an Illumina MiSeq platform. Whole exome sequencing (WES) was performed on a NextSeq 500 platform (Illumina, USA) on *n* = 217 AD, *n* = 12 DLB, *n* = 243 FTLD samples. The exons were captured by a SeqCap^®^ EZ Human Exome Probes v3.0 (Roche, Basel, Switzerland) kit with a paired-end read length of 250 bp. Whole genome sequencing (WGS) was performed on a HiSeq X platform (Illumina, USA) on *n* = 176 CTRL samples as described before [69]. Sanger sequencing was performed on a SeqStudio Genetic Analyzer (Applied Biosystems, Waltham, MA, USA) and on an AB3730 DNA analyzer (Life Technologies, Waltham, MA, USA) to confirm selected variants with a low coverage. The chromatograms were viewed through a CLC Main Workbench 20.0.4 (QIAGEN, Copenhagen, Denmark).

### 4.4. Variant Annotation, Filtering and Bioinformatics

Sequence reads from amplicon-based target enrichment were processed for quality-control purposes with the FastQC tool before alignment to the hg19 human reference sequence using Spliced Transcripts Alignment to a Reference (STAR), (The National Human Genome Research Institute, Bethesda, MD, USA) software. Duplicated reads were removed with Picard tools. Local realignment, recalibration, and variant calling were performed with the Genome Analysis Tool Kit (GATK) [70]. Variants with a QUAL < 20 and QualByDepth < 3 were excluded from the analysis. WES reads were aligned to the reference genome GRCh37 using a Burrows-Wheler Aligner (BWA) implemented using in house Genomecomb software [71] and variants were called using the GATK Haplotype Caller. WGS reads were aligned to the GRCh37 genome using Illumina’s (Illumina, San Diego, CA, USA) standard iSAAC aligner and variants were called using the iSAAC variant caller. Variant mapping to the 50 endo-lysosomal pathway genes were extracted from the WES and WGS datasets according to the coordinates of the BED files of the endo-lysosomal gene panel. WGS data were processed for quality control as described before [69]. Variants with a coverage < 20, genotype quality (GQ) < 99 and an allelic ratio >3 were excluded from the WES data. WES and WGS data were annotated using in house Genomecomb software [71]. Gene coordinates and transcripts were annotated using the RefSeq genes. Detected variants were filtered, including all “non synonymous” SNVs and premature stop codons (stop-gain, essential splice, site, frameshift indels), all variants with a Combined Annotation Dependent Depletion (CADD) score > 20, all variants annotated as damaging (D) or potentially damaging (P) according to Polymorphism Phenotyping v2 (PolyPhen2_HDIV) and with a frequency in the non-Finnish European exomes cohort reported in the Genome Aggregation Database v2.1.1. (gnomAD_NFE) < 0.01. Localization of the variants within the protein sequence was performed with Elaspic [72] and Interpro [73]. The prediction of the effect on protein stability of single amino acid substitutions was performed using two different computational tools, I-mutant 2.0 (https://folding.biofold.org/i-mutant/i-mutant2.0.html, accessed on 10 August 2021 [74]) and Mu-Pro (http://mupro.proteomics.ics.uci.edu, accessed on 10 August 2021 [75]). The change in the Gibbs free energy (ΔΔG) between wild-type and mutant protein was calculated. For both computational tools, a ΔΔG < 0 indicates a decreased protein stability upon substitution.

### 4.5. Statistical Analyses

For testing associations between rare variants and phenotypes, burden tests focusing on the cumulative effects of rare variants in genetic regions was adopted [76]. In addition, to overcome the potential loss of power when the assumption of causal and same direction effects is violated, a test which builds upon the kernel machine regression framework (sequence kernel association test—SKAT) was applied. Finally, the optimal unified test (SKAT-O), obtained as an optimal linear combination of the burden test and SKAT, was used to maximize the power [77]. For single variant tests, the efficient resampling method (ER) for a score statistic was chosen and the minimum achievable *p*-value (MAP) was provided; while, for multiple variant tests, a hybrid method, based on the total minor allele count (MAC), the number of individuals with minor alleles (m) and the degree of case-control imbalance, was adopted [77]. In addition, multiple testing false discovery rate and Bonferroni corrections were applied.

## Figures and Tables

**Figure 1 ijms-22-13633-f001:**
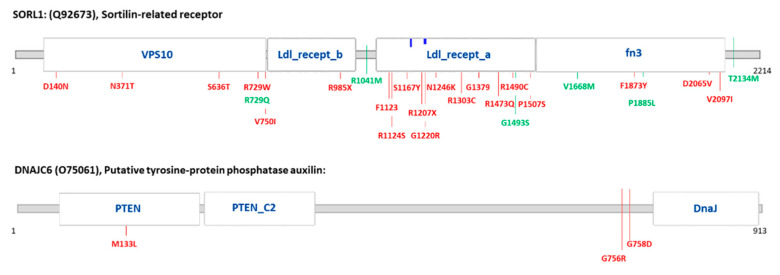
Localization of variants present in the genes of interest. Variant localization in the protein sequence, according to the Elaspic and Interpro webserver, and the relative functional domains involved. Variants present in the patient group are shown in red; Variants present in the control group are shown in green. VPS10: VPS10 domain; Ldl_recept_a: LDL receptor class a; Ldl_recept_b: LDL receptor class b; fn3: Fibronectin type 3 domain; PTEN: PTEN domain; PTEN_C2: C2 domain of PTEN tumor-suppressor protein; Dnaj: DnaJ domain.

**Table 1 ijms-22-13633-t001:** Demographic variables of patients and controls.

	AD (*n* = 282)	DLB (*n* = 114)	FTLD (*n* = 301)	CTRL (*n* = 251)	*p* Value
Sex (% Female)	**61.3**	43.9	46.8	47.4	0.0005 ^1^
Age, years	**67.0 ± 9.8**	**75.2 ± 7.6**	**67.2 ± 10.1**	**62.0 ± 9.4**	<0.0001 ^2^
Age at disease onset, years	65.0 ± 9.6	**72.5 ± 8.2**	64.4 ± 10.4	-	<0.0001 ^2^

Kolmogorov–Smirnov test was used to evaluate normality. ^1^ Chi-square test; ^2^ Kruskal–Wallis test. The groups which differed from others (post hoc tests) are reported in bold. Mean ± standard deviation.

**Table 2 ijms-22-13633-t002:** List of previously reported disease-associated variants identified in AD, DLB and FTLD.

Gene	AA Change	Variant Type	dbSNP	∆∆G Mu-Pro and I-Mutant		Diagnostic Group (Number of Carriers)	Previously Identified Diseases
**SORL1**	p.N1246K	non synonymous	rs1699102	−0.39288546	−1.30	AD (1)	AD [22,23]
	p.N371T	non synonymous	rs150609294	−1.2548181	−0.66	AD (1); FTLD (1)	AD [24]
	p.D2065V	non synonymous	rs140327834	−0.28006864	−0.57	AD (4); DLB (2); FTLD (6)	AD [24]
**DNAJC6**	p.M133L	non synonymous	rs61757223	n.a.	n.a.	AD (1); DLB (1)	PD [25]

AA, amino acid; dbSNP, database of single nucleotide polymorphism; ΔΔG, protein stability free energy change.

**Table 3 ijms-22-13633-t003:** Predicted null variant in genes where LOF is a known mechanism of the disease.

Gene	AA Change	Variant Type	dbSNP	Diagnostic Group (Number of Carriers)	Previously Associated Diseases (LOF Mechanism)
PPT1	p.R48X	LOF	-	AD (1)	NCL [30,31]
SORL1	p.R985X	LOF	rs372188860	AD (1)	AD [26,27,28,29]
	p.R1207X	LOF	rs774626685	AD (1)	

AA, amino acid; LOF, loss of function variant; dbSNP, database of single nucleotide polymorphism; NCL, neuronal ceroid lipofuscinosis.

**Table 4 ijms-22-13633-t004:** List and clinical characteristics of previously reported disease-associated variants and predicted null variants.

Gene	Function *	AA Change	Variant Type	Diagnostic Group	Disease Onset	Family History	Association with Neurodegenerative Diseases in the Literature
SORL1	Intracellular protein sorting/transport	N1246K	non synonymous	AD	72	F	AD—Genetic and molecular studies in humans [22,23,24,32] FTLD—Genetic studies in humans [33] PD—Genetic studies in humans [34]
		N371T	non synonymous	AD	53	U	
				FTLD (bvFTD)	72	U	
		D2065V	non synonymous	AD	60	F	
				AD	61	F	
				AD	50	AS	
				AD	66	F	
				DLB	74	AS	
				DLB	68	AS	
				FTLD (bvFTD)	63	AS	
				FTLD (bvFTD)	75	F	
				FTLD	62	U	
				FTLD (PPA)	68	F	
				FTLD (bvFTD)	76	F	
				FTLD (PPA)	71	AS	
		R985X	LOF	AD	46	F	
		R1207X	LOF	AD	68	U	
DNAJC6	Uncoating of clathrin-coated vesicles	M133L	non synonymous	AD	65	U	PD—Genetic studies in humans [25,35,36]
				DLB	67	U	
PPT1	Catabolism of lipid-modified proteins during lysosomal degradation	R48X	LOF	AD	61	F	AD—Molecular studies in animal models [37] FTLD—Molecular studies in animal models [38]

AA, amino acid; LOF, loss of function; bvFTD, behavioral variant FTD; PPA, primary progressive aphasia; F, positive family history; AS, apparently sporadic; U, unknown; * For a complete list of gene functions reported by Gene Ontology see Appendix A.

**Table 5 ijms-22-13633-t005:** Single variant analysis.

Gene	Variant	gnomAD_NFE	CADD	Poly Phen2	∆∆GMu-Pro and I-Mutant		Diagnostic Group	*p* Value	MAP	*p* Value Fdr	*p* Value Bonf.
**SORL1**	** p.D2065V **	0.00416 ^a^	28.5	D	−0.28	−0.57	AD + DLB + FTLD	**0.031**	<0.001	0.611	1
							DLB	**0.048**	0.048	0.618	1
							FTLD	**0.021**	0.004	0.656	1
**AGRN**	p.V554M	0.0065 ^a^	25.8	D	−0.52	0.03	DLB	0.048	0.048	0.618	1
**NEU1**	p.R397W	0.00004617 ^b^	26.8	D	−0.95	−0.76	DLB	0.048	0.048	0.618	1
**TOM1**	p.V67A	0.00007169 ^b^	26.5	D	−1.165	−2.26	DLB	0.048	0.048	0.618	1
**ABCA2**	p.H1449P	0 ^c^	24.3	D	n.a.	n.a.	DLB	0.048	0.048	0.618	1

gnomAD_NFE, genome aggregation database non-Finnish European; ^a^, 0.001 < gnomAD_NFE ≤ 0.01; ^b^, 0 < gnomAD_NFE ≤ 0.001; ^c^, gnomAD_NFE = 0; CADD, combined annotation dependent depletion; PolyPhen2, polymorphism phenotyping v2; D, damaging; P, potentially damaging; ΔΔG, protein stability free energy change; MAP, minimum achievable *p*-value; Fdr, false discovery rate corrected; Bonf., Bonferroni corrected. Previously reported variants are underlined in bold.

**Table 6 ijms-22-13633-t006:** Gene burden analysis.

Gene	Diagnostic Group	Number of Variants	Number of Carriers	*p* Value Burden	*p* Value Skato	*p* Value Skat	*p* Value Fdr	*p* Value Bonf.
**SORL1**	AD + DLB + FTLD	26	40	0.127	0.076	**0.038**	0.961	1
	FTLD	19	26	**0.025**	**0.016**	**0.017**	0.481	0.963
**DNAJC6**	DLB	2	2	**0.048**	**0.048**	**0.048**	0.776	1
**NEU1**	DLB	1	2	0.048	0.048	0.048	0.776	1
**AP2A2**	AD + DLB + FTLD	7	18	0.037	0.058	0.392	0.522	1
	AD	4	8	0.043	0.090	0.364	0.622	1
	FTLD	5	9	0.022	0.080	0.374	0.481	0.858

Fdr, false discovery rate corrected; Bonf., Bonferroni corrected.

**Table 7 ijms-22-13633-t007:** List and clinical characteristics of multiple variant carriers.

Patient	Diagnostic Group	FTLD Subtype	Disease Onset	FH	Sex	Study Group	Variants		
1	FTLD	PPA	68	F	M	Belgium	** SORL1 p.D2065V ^a^ **	GGA3 p.K99R ^a^	-
2	AD	-	68	U	M	Belgium	** SORL1 p.R1207X ^b^ **	SORL1 p.D140N ^b^	-
3	AD	-	65	U	M	Belgium	** DNAJC6 p.M133L ^b^ **	AGRN p.A897V ^a^	-
4	AD	-	61	F	F	Belgium	** PPT1 ** ** p.R48X ^b^ **	GNPTG p.R66Q ^b^	**-**
5	FTLD	PPA	57	F (low) *	F	Italy	SORL1 p.S1167Y ^b^	ABCA2 p.H1449P ^c^	GPC1 p.R90W ^b^
6	FTLD	PPA	63	AS	F	Italy	SORL1 p.R729W ^b^	CTSA p.P330A ^b^	GGA3 p.P235L ^b^
7	FTLD	bvFTD	82	F (low) *	M	Italy	SORL1 V2097I ^a^	ABCA2 S1378F ^b^	-
8	FTLD	-	66	F	M	Belgium	SORL1 p.S636T ^a^	VPS39 p.V473M ^b^	
9	FTLD	-	49	AS	F	Belgium	VPS52 p.Y508C ^b^	VPS52 p.R578W ^b^	-
10	FTLD	bvFTD + IBM	70	F	M	Belgium	AGRN p.R956H ^b^	CD81 p.G129R ^b^	HGS p.L525V ^b^
11	FTLD	PPA	50	F	M	Italy	ABCA2 p.H1449P ^c^	ATP6V0D1 c.C817-2A ^c^	**-**
12	FTLD	bvFTD	59	F (medium) *	M	Italy	ABCA2 p.H1449P ^c^	GGA2 p.L83I ^b^	**-**
13	FTLD	PPA	60	F (high) *	M	Italy	AGRN p.V1691M ^b^	ATP6V0D1 c.C817-2A ^c^	-
14	FTLD	PPA	54	F (high) *	M	Italy	DNM2 p.R318W ^b^	ATP6V0D1 c.C817-2A ^c^	-
15	DLB	-	70	F	M	Italy	GGA2 p.S39W ^c^	GGA3 p.P40R ^b^	**-**
16	FTLD	bvFTD	65	F	M	Belgium	GGA2 p.R105G ^b^	VPS39 p.F573L ^b^	**-**
17	FTLD	bvFTD	39	AS	F	Belgium	GNPTG p.R186W ^c^	MGRN1 p.P67L ^c^	**-**
18	DLB	-	78	F	M	Belgium	NEU1 p.R397W ^b^	TOM1 p.V67A ^b^	**-**

bvFTD, behavioral variant FTD; PPA, primary progressive aphasia; IBM, inclusion body myopathy; FH, family history; F, positive family history; AS, apparently sporadic; U, unknown; * Fostinelli et al., 2018; ^a^, 0.001 < gnomAD_NFE ≤ 0.01; ^b^, 0 < gnomAD_NFE ≤ 0.001; ^c^, gnomAD_NFE = 0. Previously reported variants and LOF variants are in underlined bold text.

## Data Availability

The raw data supporting the conclusions of this article will be made available by the corresponding author upon request.

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
