# Peer review of "Investigating the Endo-Lysosomal System in Major Neurocognitive Disorders Due to Alzheimer’s Disease, Frontotemporal Lobar Degeneration and Lewy Body Disease: Evidence for SORL1 as a Cross-Disease Gene"

_ijms, 2021, doi:10.3390/ijms222413633_

Round 1
Reviewer 1 Report
This is a well written, interesting manuscript, that will contribute to the literature.
Reviewer 2 Report
The manuscript studied the role of Endo-Lysosomal System in three major neurodegenerative diseases leading to dementia.
My comments
-Introduction: the sentences are too long and confusing and paragraphs are too long also
-Materials and methods: what were exclusion criteria for patients. Also at which stage was the samples obtained
-The conclusion should be consistent with the aim. the authors mentioned in the aim that they did gene sequencing. So, in conclusion they should mention genes controlling intracellular protein sorting/transport, the clathrin uncoating of vesicles and the regulation of enzymatic activity in lysosomes, because the genes were assessed not the proteins or enzyme activity
Reviewer 3 Report
13 December 2021
Review on the manuscript titled “Investigating the Endo-Lysosomal System in Alzheimer’s Disease, Frontotemporal Lobar Degeneration and Dementia with Lewy Bodies: evidence for SORL1 as cross-disease gene” by Benussi L et al., submitted to International Journal of Molecular Sciences (IJMS).
Manuscript ID: ijms-1505336
Dear Authors,
The dysfunction of the endo-lysosomal system has been linked to neurodegeneration in patients with major neurocognitive disorders. The authors studied the sequences of 50 endo-lysosomal pathway genes in brain samples of patients with Alzheimer’s disease (AD), dementia with Lewy bodies (DLBs), or frontotemporal lobar degeneration (FTLD). The results showed that the gene variants of the endo-lysosomal pathway are linked to major neurocognitive disorders. The authors concluded that the endo-lysosomal pathway, especially gene SOL1 plays an important role in neurodegenerative dementia
Please reconsider the following parts:
- A graphic abstract summarizing the manuscript is highly recommended.
- The title and the body of the manuscript: “Dementia” is a derogatory term. Please use “major neurocognitive disorders”.
- Page 1, Abstract:
- Please define abbreviations in the first appearance.
- Please clearly and proportionally present background, purpose, methods, results, and conclusion. Background may include the previous findings and conclusion may include the significance of this study.
- Page 2, Introduction:
- Please separate the section into three to four paragraphs.
- 95% of major neurocognitive disorders are caused by AD, stroke-induced neurodegeneration (SND), and DLBs. Please present a rationale to exclude SND and include FTLD.
- The pathogenesis of neurodegenerative diseases is reviewed recently.
- Please present a rationale to choose 50 genes and a table or figure showing their functions in the endo-lysosomal system.
- Suggested references: https://doi.org/10.1176/appi.books.9780890425596; doi: 10.1007/s00702-020-02232-9; https://www.mdpi.com/1422-0067/21/7/2431; doi: 10.3390/ijms21197332.
- Pages 3-10, Results: Please present a table or figure summarizing the function of genes studied, variant type, and links to cognitive function, among others.
- Pages 9,10, Discussion: Present the previous data, significance of current study, more weaknesses or limitation in the present study, potentials, the ultimate goal, research or knowledge needed to achieve, the biggest challenge in this goal, and future research direction, among others
The manuscript contains one figure, six tables, and 65 references. The manuscript carries important value presenting the link between the gene variants of the endo-lysosomal system and major neurocognitive disorders. I recommend this manuscript for publication after major revision.
I declare no conflict of interest regarding this manuscript.
Round 2
Reviewer 2 Report
The authors addressed my comments
Reviewer 3 Report
16 December 2021
Review on the manuscript titled “Investigating the Endo-Lysosomal System in Alzheimer’s Disease, Frontotemporal Lobar Degeneration and Dementia with Lewy Bodies: evidence for SORL1 as cross-disease gene” by Benussi L et al., submitted to International Journal of Molecular Sciences (IJMS).
Manuscript ID: ijms-1505336
Dear Authors,
The dysfunction of the endo-lysosomal system has been linked to neurodegeneration in patients with major neurocognitive disorders. The authors studied the sequences of 50 endo-lysosomal pathway genes in brain samples of patients with Alzheimer’s disease (AD), dementia with Lewy bodies (DLBs), or frontotemporal lobar degeneration (FTLD). The results showed that the gene variants of the endo-lysosomal pathway are linked to major neurocognitive disorders. The authors concluded that the endo-lysosomal pathway, especially gene SOL1 plays an important role in neurodegenerative dementia. The authors addressed their response properly and the manuscript is revised accordingly. The manuscript contains one figure, seven tables, and 77 references. The manuscript carries important value presenting the link between the gene variants of the endo-lysosomal system and major neurocognitive disorders. I recommend this manuscript for publication in current form.
I declare no conflict of interest regarding this manuscript.
Best regards,
Masaru Tanaka, M.D., Ph.D.